# A Comprehensive Design of Six-Axis Force/Moment Sensor

**DOI:** 10.3390/s21134498

**Published:** 2021-06-30

**Authors:** Anton Royanto Ahmad, Terrence Wynn, Chyi-Yeu Lin

**Affiliations:** 1Department of Mechanical Engineering, National Taiwan University of Science and Technology, Taipei 106, Taiwan; D10703814@mail.ntust.edu.tw (A.R.A.); B10835032@mail.ntust.edu.tw (T.W.); 2Center for Cyber-Physical System, National Taiwan University of Science and Technology, Taipei 106, Taiwan; 3Taiwan Building Technology Center, National Taiwan University of Science and Technology, Taipei 106, Taiwan

**Keywords:** six-axis force/moment sensors, strain gage, data acquisition board

## Abstract

Strain gage type six-axis force/moment (F/M) sensors have been largely studied and implemented in industrial applications by using an external data acquisition board (DAQ). The use of external DAQs will ill-affect accuracy and crosstalk due to the possibility of voltage drop through the wire length. The most recent research incorporated DAQ within a relatively small F/M sensor, but only for sensors of the capacitance and optical types. This research establishes the integration of a high-efficiency DAQ on six-axis F/M sensor with a revolutionary arrangement of 32 strain gages. The updated structural design was optimized using the sequential quadratic programming method and validated using Finite Element Analysis (FEA). A new, integrated DAQ system was designed, tested, and compared to commercial DAQ systems. The proposed six-axis F/M sensor was examined with the calibrated jig. The results show that the measurement error and crosstalk have been significantly reduced to 1.15% and 0.68%, respectively, the best published combination at this moment.

## 1. Introduction

Until now, the six-axis force/moment (F/M) sensor has been extensively investigated and has been developed with many types. F/M sensors can be classified based on the transducer used or the structure type. As objectively stated in review studies [1,2], each has some advantages and disadvantages. In addition, F/M sensors can be classified based on the decoupling method used for decoupling of the force sensor signal, such as linear regression [3], the least squares method (LSM) [4], least squares support vector machines (LS-SVMs) [5], neural networks [6], the shape–form–motion approach [7], and support vector regression (SVRs) [8,9].

Normally, a commercial data acquisition board (DAQ) will be used for acquiring data from a six-axis F/M sensor producing analog output signals. The use of external DAQ will deteriorate accuracy and crosstalk due to the possibility of voltage drop through wire length. For industrial applications which require high precision, this becomes a major drawback. Recently, a few researchers have constructed their own DAQ within a six-axis F/M sensor with a capacitance transducer [10,11] and an optical transducer [12]. However, in the field of strain gage type F/M sensors, researchers [4,13] made this possible while having a relatively large sensor dimension. ATI^®^ made an accurate strain gage type F/M sensor with an embedded DAQ; however, this sensor was still prone to temperature noises. 

The novelties of this study are the simultaneous implementation of a novel structural design, a new strain gage arrangement, and the efficient integrated DAQ design to achieve low-cost production whilst also creating an accurate six-axis F/M sensor. The Sequential Quadratic Programming (SQP) method on Timoshenko’s strain equation was used to optimize the structural design with a fast and accurate result. Two types of strain gage arrangements that help to attain immunity to temperature noise were analyzed using Finite Element Analysis (FEA) and validated through experiments. The integrated DAQ was designed, tested, and compared to a commercial external DAQ to ensure that it meets the specifications. This research also provides validation for the calibration/testing jig by using a commercial force sensor, the ATI AXIA 80. Finally, the decoupling method was realized based on a simple LSM. 

The content of this paper is divided into seven major sections. Following the introduction of the study’s objective in Section 1, the structural design of this sensor will be discussed in Section 2. The specification, strain gage allocation analysis, design optimization, and accuracy and crosstalk of both 24 and 32 strain gage arrangements are all discussed. Section 3 discusses the design of the data acquisition board, beginning with specifications and moving on to the IC selection and PCB layout. Section 4 will discuss the calibration and testing methods used to study the performance of this force sensor. Section 5 and Section 6 present and discuss the results of the force sensors based on either 24 or 32 strain gage arrangements, using either commercial data acquisition hardware or a custom-made data acquisition board. Section 7 will contain the conclusions. 

## 2. Structural Design

### 2.1. Specification of the Sensor

The novel six-axis F/M structural design is based on the design specifications shown in Table 1. As shown in Figure 1a, the six-axis F/M sensor is made up of three major components: an elastic body, a top flange, and a bottom flange. The sensor structure is constructed of aluminum 7075-T6, which has a Young’s modulus of 71.7 GPa and a tensile yield strength of 503 MPa. Figure 1b depicts the elastic body design of a six-axis F/M sensor. The elastic body is made up of four elastic beams connected by a square platform in the center and has four elastic boundaries. The tool is attached to the top flange, which transfers forces and moments to the elastic body via a square platform. The bottom flange serves two purposes: as an end cap and as a support for the elastic body. 

### 2.2. Strain Gage Allocation Analysis

The strain gages are allocated based on three criteria: the maximum strain, isotropy of the material involved, and non-linearity prevention [14]. To begin the investigation, the location where the strain gage would be attached was chosen, which was the surface of the elastic beam. Second, ANSYS, the FEA software, was used to analyze the normal strain and linearity in the middle line of the elastic beam, as shown in Figure 2. Third, the maximum load of the force or moment was applied solely to the elastic body’s square platform. Finally, as shown in Figure 3, the elastic beam’s normal strain was plotted in the *X*-axis direction. Due to the symmetry of the structure, the normal strain was similar in both the X and Y directions. The strain was non-linear near both ends of the elastic beam, as shown in Figure 3. This non-linearity occurred because of the fillet structure on both tails that reduces the stress concentration [15]. To achieve a high strain while avoiding non-linearity, all strain gages needed to be spaced 5 mm apart from the elastic body’s square platform.

### 2.3. Strucutral Design Optimization

To meet the design specification, the initial design of an elastic body, as shown in Figure 4 and Table 2, was used to implement an optimization process. The fast and accurate optimization method used in this research was SQP on Timoshenko’s strain equation [15,16,17]. These equations, Equations (1)–(6), were derived from the study [15]. The following forces were used in the optimization: F_X_, F_Y_, F_Z_ = 500 N and M_X_, M_Y_, M_Z_ = 20 Nm. To protect the strain gage in an elastic state, the following specific strain limits were set for each axis: $\varepsilon_{F_{X}}=\varepsilon_{F_{Y}}=\varepsilon_{M_{X}}=\varepsilon_{M_{Y}}=500,\;\varepsilon_{F_{Z}}=\varepsilon_{M_{Z}}=250$.
(1)$$\varepsilon_{F_{X}}=\frac{zN\;F_{x}\left(c-x\right)}{E\;I_{AX}\;\left(4M+2N\right)}\;;\;\;M=\frac{c}{kG\;A_{AX}}+\frac{c^{3}}{3\;E\;I_{AX}};\;\;N=\frac{e}{2kG\;A_{GX}}+\frac{e^{3}}{96\;E\;I_{GX}}$$
(2)$$\varepsilon_{F_{Z}}=z\;F_{AZ}\frac{\left(c-x\right)}{E\;I_{AZ}}=\frac{z\;F_{Z}\;\left(c-x\right)}{4E\;I_{AZ}}\;;\;F_{Z}=4F_{AZ}$$
(3)$$\varepsilon_{M_{X}}=\frac{z\left(c-x\right)c\left(c+\frac{g}{2}\right)M_{X}}{E\;I_{AX}\;\left[G\;I_{AM_{x}}\left(2P+Q\right)+2c{\left(c+\frac{g}{2}\right)}^{2}\right]};P=\frac{c}{kG\;A_{AZ}}+\frac{c^{3}}{3\;E\;I_{AZ}};Q=\frac{e}{2kG\;A_{EZ}}+\frac{e^{3}}{96\;E\;I_{EZ}}$$
(4)$$\varepsilon_{M_{Z}}=\frac{z\;M_{Z}\;\left(c-x\right)}{\left(4c+2g\right)E\;I_{AZ}}$$
(5)$$I_{AX}=\frac{a^{3}b}{12}\;;I_{GX}=\frac{d^{3}f}{12}\;;I_{AZ}=\frac{b^{3}a}{12}\;;I_{EZ}=\frac{f^{3}d}{12}\;;I_{AM_{X}}=\beta a^{3}b\;,\;\beta =0.141$$
(6)$$k=\frac{10\left(1+\eta \right)}{\left(12+11\eta \right)};\;G=\frac{E}{2\;\left(1+\eta \right)}$$

The following abbreviations have the following meanings: *a* = width of the elastic beam, *b* = height of the elastic beam, *c* = length of the elastic beam, *d* = width of the elastic boundary, *e* = length of the elastic boundary, *f* = height of the elastic boundary, *g* = width of the square platform, *x* = location of the strain gage, and *z* = half-width/height of the elastic beam.

The optimization was carried out using MATLAB. Depending on the loading, the results were different, as shown in Figure 5 and Table 3. This optimization’s objectives were to minimize Equations (7)–(12). The criterion for stopping was a step size of less than 10^−6^. The constraints are defined as follows: 8.0≤a,b≤9.0, 10.0≤c≤20.0, 0.8≤d≤1.0, 20.0≤e≤35.0, 10.0≤f≤15.0, 20.0≤g≤35.0.
(7)$$\varepsilon_{F_{X}}-500$$
(8)$$\varepsilon_{F_{Z}}-250$$
(9)$$\varepsilon_{M_{X}}-500$$
(10)$$\varepsilon_{M_{Z}}-250$$
(11)$$\varepsilon_{F_{X}}+\varepsilon_{M_{Z}}-750$$
(12)$$\varepsilon_{F_{Z}}+\varepsilon_{M_{X}}-750$$

When the structure was subjected to a load of F_X_ or F_Y_ = 500 N, the objective function was to minimize Equation (7). The number of iterations required to meet the stopping criteria was eight. Table 3 shows the optimized parameter of the elastic body under F_X_ loading, which has a function value of 9.94 × 10^−5^. When the structure was subjected to a load of F_Z_ = 500 N, the objective function was to minimize Equation (8). The number of iterations required to meet the stopping criteria was seventeen. Table 3 shows the optimized parameter of the elastic body under F_Z_ loading, which has a function value of 4.54 × 10^−4^. When the structure was subjected to a load of M_X_ or M_Y_ = 20 Nm, the objective function was to minimize Equation (9). The number of iterations required to meet the stopping criteria was seventeen. Table 3 shows the optimized parameter of the elastic body under M_X_ loading, which has a function value of 9.81 × 10^−4^. When the structure was subjected to a load of M_Z_ = 20 Nm, the objective function was to minimize Equation (10). The number of iterations required to meet the stopping criteria was fourteen. Table 3 shows the optimized parameter of the elastic body under M_Z_ loading, which has a function value of 2.22 × 10^−4^.

However, the design of the structure needed to consider multiple loads at a time. The strain gage itself had an allowable strain of up to 50.000 µm/m before breaking, but to obtain a longer life cycle more than 10^7^, the strain had to be kept below 1000 µm/m. For performing a pure load, 500 µm/m became the objective strain in optimization. For multiple loads, the objective strain could be up to 750 µm/m. When the structure was subjected to multiple loads of F_X_ or F_Y_ = 500 N and M_Z_ = 20 Nm, the objective function was to minimize Equation (11). The number of iterations required to meet the stopping criteria was 31. Table 3 shows the optimized parameter of the elastic body under F_X_ + M_Z_ loading, which has a function value of 1.82 × 10^−3^. When the structure was subjected to multiple loads of F_Z_ = 500 N and M_X_ or M_Y_ = 20 Nm, the objective function was to minimize Equation (12). The number of iterations required to meet the stopping criteria was 41. Table 3 shows the optimized parameter of the elastic body under F_Z_ + M_X_ loading, which has a function value of 2.65 × 10^−3^. 

The average value of all parameters is the value that would be used for the manufacturing process, with tolerance based on minimum and maximum values of all optimized parameters. The prototype was the actual size of the manufactured structure. Even it was not the same as the average value, it would still have been within the tolerances. The rest of simulation process was based on the prototype size so that it could be compared fairly. 

The strain analysis of the elastic beams required during optimization must be performed using the finite element analysis, as shown in Figure 6, and Timoshenko equation. As shown in Figure 7, an analytical solution derived from Timoshenko equation is presented alongside the results of the numerical solution derived from ANSYS. These results show that the Timoshenko equation produces results that are similar to numerical results. The final tail of the graph for the numerical results, on the other hand, shows non-linear strains, whereas the graph for the Timoshenko equation results only shows a straight line. These findings are consistent with those of previous studies that compared Timoshenko results to numerical results [18,19]. An interesting finding was that when the structure was experiencing multiple loads (F_X_ = 500 N, F_Y_ = 500 N, F_Z_ = 500 N, M_X_ = 20 Nm, M_Y_ = 20 Nm, and M_Z_ = 20 Nm), the normal strain on middle side line, Figure 2e, was the same as the strain based on Equation (1) + Equation (4), and the normal strain on the middle top line, Figure 2f, was the same as the strain based on Equation (2) + Equation (3). This is an indication that this structure is a mechanically decoupled structure because the normal strain on the middle side line would not be affected by the forces that affect the normal strain on middle top line and vice versa. 

The placing of the strain gage location on elastic beams at a distance of 5 mm from the square platform was the focus of this study. Table 4 compares the numerical simulation results to the analytical solutions at 5 mm from the square platform. However, the current study’s result is preferable when compared to the 5.47% deviation found in some previous studies [14,20]. This is due to the fact that the Timoshenko equation is suitable for estimating the optimum design of the mechanically decoupled elastic cross-beam structure [21].

### 2.4. Accuracy and Crosstalk

Using Equation (13), this structure was then tested to quantify the crosstalk. A comparison of two types of strain gage arrangements was presented in this study. First, 24 strain gages were used, followed by 32 strain gages. Both of these arrangements used the full Wheatstone bridge to anticipate temperature noise.
(13)$$C_{\mathrm{ij}}=\left(\frac{S_{\mathrm{ij}}}{S_{\mathrm{ii}}}\right)\times 100\%$$

S_ij_ is the measured normal strain, with i = 1, …, 6 representing the measurement provided by the strain gage bridge and j = 1, …, 6 representing the specified load on the force sensor. For example, S_11_ represents the reading provided by the strain gage bridge *S*_FX_ when F_X_ is applied to the force sensor. C_ij_ is the crosstalk that occurs when i ≠ j. For example, C_12_ indicates the reading provided by *S*_FX_ when F_X_ is applied.

#### 2.4.1. 24 Strain Gage Arrangement 

The strain gage arrangement used in this study is similar to the one used in [20,22], as shown in Figure 8. The blue font signifies the visible strain gages, and the red font denotes the invisible strain gages (opposite side of the beam).
(14)$$S_{\mathrm{Fx}}=\left(\frac{1}{4}\right)\left[\left(R_{28}-R_{31}\right)+(R_{15}-R_{12}\right)]$$
(15)$$S_{\mathrm{Fy}}=\left(\frac{1}{4}\right)\left[\left(R_{8}-R_{3}\right)+(R_{19}-R_{24}\right)]$$
(16)$$S_{\mathrm{Fz}}=\left(\frac{1}{4}\right)\left[\left(R_{1}-R_{6}\right)+(R_{18}-R_{21}\right)]$$
(17)$$S_{\mathrm{Mx}}=\left(\frac{1}{4}\right)\left[\left(R_{25}-R_{30}\right)+(R_{13}-R_{10}\right)]$$
(18)$$S_{\mathrm{My}}=\left(\frac{1}{4}\right)\left[\left(R_{17}-R_{22}\right)+(R_{5}-R_{2}\right)]$$
(19)$$S_{\mathrm{Mz}}=\left(\frac{1}{4}\right)\left[\left(R_{32}-R_{27}\right)+(R_{16}-R_{11}\right)]$$

As shown in Table 5, the highest crosstalk produced by this arrangement is −0.34%. The strain gage arrangement can differentiate the loads into six-axis forces and moments when executing the multiple loading on the structure. In practice, however, it is extremely difficult to connect two single strain gages in parallel. As a result, the strain gage used in this study is a double parallel strain gage from HBM, which simplifies the process of attaching the strain gage. To comply with this arrangement, 16 double strain gages were used on this sensor, 4 for each elastic beam.

#### 2.4.2. 32 Strain Gage Arrangement

The sensor has 32 strain gages because of the usage of 16 double parallel strain gages. Therefore, eight more strain gages could be utilized. From the FEA result, it can be seen that, especially when the structure was subjected to F_Z_ or M_Z_ loading, four elastic beams were affected by that force. Furthermore, in a 24 strain gage arrangement, the Wheatstone bridges for *S*_FZ_ and *S*_MZ_ were only supported by two beams. The eight additional strain gages could be added to the *S*_FZ_ and *S*_MZ_ bridges, as shown in Equations (20) and (21).
(20)$$S_{\mathrm{Fz}}=\left(\frac{1}{8}\right)\left[\left(R_{1}-R_{6}\right)+(R_{18}-R_{21}\right)+\left(R_{26}-R_{29}\right)+(R_{9}-R_{14})]$$
(21)$$S_{\mathrm{Mz}}=\left(\frac{1}{8}\right)\left[\left(R_{32}-R_{27}\right)+(R_{16}-R_{11}\right)+\left(R_{7}-R_{4}\right)+(R_{23}-R_{20})]$$

This arrangement is now a 32-strain gage arrangement. Table 6 depicts the crosstalk for this configuration. The crosstalk was slightly better for *S*_FZ_ and *S*_MZ_, but it was not possible to reduce the maximum crosstalk that occurred on *S*_FX_ while the structure was loaded with M_Y_, which was −0.34%. This was because of the changes only occurring on *S*_FZ_ and *S*_MZ_. Meanwhile, this result outperforms, by less than 0.51%, the simulation results of studies by other researchers that have similar specifications and used the strain gage as a transducer [20,22,23,24,25]. When performing multiple loading on the structure, the strain gage arrangement could differentiate the loads into six-axis forces and moments. 

## 3. Integrated Data Acquisition Board (DAQ)

To build an integrated DAQ, the electronic specification for this sensor needed to be defined first, as shown in Table 7. The resolution and the sampling rate of this sensor govern the selection of all electronic components within the PCB. The maximum range for F_X_, F_Y_ and F_Z_ are ± 500 N, with a resolution of ± 0.1 N. According to Equation (12), the resolution bit for ADC is 13.28. The maximum range for M_X_, M_Y_, M_Z_ are ± 20 Nm, with a resolution of ± 0.001 Nm. According to equation (22), the resolution bit for ADC is 14.28. The effective bit resolution of ADC should be examined in the datasheet. The ADC bit resolution differs from the effective bit resolution. At the same ADC bit resolution, different sampling rates, gain amplifiers, and filtering setups might result in different effective bit resolutions. Based on this consideration, the TI ADS 124S08, a 24-bit ADC with a maximum sampling rate of 100 Hz, was chosen as an analog to digital converter for six strain gage bridges. After selecting the ADC, the microcontroller unit that will communicate with it via SPI needed to be chosen. Other factors to consider when choosing an MCU include computing speed, language, digital voltage, and so on. ATMEGA328p from ATMEL was chosen as the MCU in this research. It is an AVR MCU that can be programmed using the Arduino IDE, which is well-known and simple to use.
(22)$$bit\;ADC=\log_{2}\frac{FR}{Resolution}+1$$

The data acquisition board, as shown in Figure 9, supports two types of communication: RS485 and USB. RS485 is a serial communication protocol that is widely used in the industrial world, while USB is intended to program the MCU. In addition, as is common in the academic world, USB can be used to acquire data. The ADC will convert the signal from the strain gage bridges (Figure 10), and the MCU will collect the data from the ADC. 

Figure 11 depicts the PCB layout. The board was split into four sections. The division was made to reduce crosstalk between the digital and analog signals. The yellow bracket contains the communication section, which is made up of two ICs: USB and RS485. The blue bracket represents the data converter section, which includes the MCU and ADC. The red bracket dwellings stand for the power supply section, which contains four voltage regulators. To power the six strain gage bridges, a voltage regulator with a high accuracy was required. The final section is the Wheatstone bridge section, Figure 11b, which connects strain gages to the data acquisition section via another PCB board.

## 4. Methodology

### 4.1. Validation of the Calibration Jig

A six-axis calibration jig was developed from previous studies to apply pure forces and moments on different axes. The calibration jig was validated using a commercial F/M sensor, ATI AXIA 80, as shown in Figure 12. Some standards have already certified this six-axis F/M sensor. The sensor was attached to the calibration jig and given some loads for each axis. The weight plates utilized to load this sensor would also be utilized to load the proposed six-axis F/M sensor. The force and moment readings from this sensor were used as a benchmark, as shown in Table 8. 

### 4.2. Data Acquiring Method

Two different DAQs were used in this study to acquire sensor signals. The output voltage signal responses of all six full-bridge circuits were first measured directly using a NI-DAQ from National Instruments with PCI−6229 as a multifunctional data acquisition card. The second method was to incorporate our DAQ into the sensor, as stated in the preceding chapter. 

Experimental data were obtained via the data acquisition system during the experimental test of the decoupled six-axis F/M sensor. The experimental test was similar to the data collection process for calibration. These procedures were similar to those described in the literature [20,22].

The output voltage of the integrated DAQ, based on the calibration process, can be used to make linear equations by means of the LSM. The linear equations could be integrated with the custom DAQ. The output voltages are shown in Figure 13. As shown in Figure 13a,b, there were *S*_MY_ readings on F_X_ loadings and *S*_MX_ readings on F_Y_ loadings. These were expected to happen because of the gap between the top flange of F/M sensor to the force axis of the calibration jig. This gap represents the coupling flange thickness. The gap was 9 mm in the z-direction, so when the positive F_X_ loading was applied, it would create a positive M_Y_ loading. Likewise, if a negative F_X_ loading was applied, it would create a negative M_Y_ loading. In addition, when a positive F_Y_ loading was applied, it would create a negative M_X_ loading. If a negative F_Y_ loading was applied, it would create a positive M_X_ loading.

## 5. Results

### 5.1. Commercial DAQ Result

In this section, we will demonstrate the performance of a six-axis force sensor with two different strain gage arrangements. The data acquisition hardware used in this section is from NI-DAQ. 

#### 5.1.1. 24 Strain Gage Arrangement 

Table 9 displays the force sensor’s average measurement error and crosstalk. This equation was used to obtain the measurement error and crosstalk:
(23)$$\mathrm{error}\left(\%\right)=\left|\frac{F_{S}-F_{B}}{F_{\mathrm{SO}}}\right|\times 100\%$$
F_S_ is the measurement from the proposed F/M sensor; F_B_ is equal to the benchmark reading, based on Table 8, for calculating absolute measurement error; F_B_ is equal to zero for calculating crosstalk; and F_SO_ is the full-scale range of the proposed F/M sensor. The average measurement error illustrates the deviation of the designated bridges (the strain bridge axis is the same as the loading, i.e., *S*_FX_ with the load F_X_) of the proposed F/M from the benchmark. The crosstalk shows the deviation of the undesignated bridges (the strain bridge axis is different from the loading i.e., *S*_FX_ with the load M_Z_) of the proposed F/M from zero in percentage. The average measurement error of each bridge was shown diagonally. On *S*_MZ_, the maximum average measurement error occurred, which was 2.40%. Crosstalk was observed on *S*_MZ_, when the sensor was subjected to the M_X_ load, at a level of 3.35%. This result is unsatisfactory because it contains more than the target of 2.0%, which was the best achieved so far among researchers.

#### 5.1.2. 32 Strain Gage Arrangement 

The results in Table 10 are for a 32 strain gage arrangement with NI-DAQ. The average measurement error and crosstalk have been significantly reduced and meet our target performance criteria. The maximum measurement error on *S*_MY_ is 1.13%, and the maximum crosstalk on *S*_MX_ is 1.20%.

### 5.2. Integrated DAQ Result

After we finished calibrating and testing the sensor, we were able to obtain outstanding average measurement error and crosstalk values. In this section, we only show the 32 strain gage arrangement based on the previous step. We created the PCB for the 32 strain gage arrangement, as discussed in Section 3. Table 11 shows that the maximum average measurement error occurs on *S*_MY_, 1.15%, which is comparable to the values that would be obtained using a commercial DAQ. In terms of crosstalk, we were able to achieve an even better result of 0.68%.

## 6. Discussion

The overall goal of this research is to create a six-axis force sensor with the specifications listed in Table 1 that performs similarly to, or even better than, commercial six-axis force sensors while costing less. The process began with the structure being optimized using the SQP method on Timoshenko’s strain equation. We evaluated the placement of the strain gage on the force sensor using the 24 strain gage arrangement based on [20,22]. To study crosstalk, we simulated the combination of the optimized structure and the unique strain gage arrangement. Following that, we built the force sensor. We performed calibration and testing on this combination, and the results did not meet the criteria. 

To improve the outcome, we started with the cheapest option, which included a waste strain gage. We experimented with different strain gage arrangements. Instead of a single strain gage per piece, we used a double parallel strain gage, which included two strain gages per piece. It was expected to make the placement process easier and more accurate. However, eight strain gages were not used in the 24 strain gage arrangement. One-piece had two strain gages, and one force sensor was made up of 16 pieces. As a result, the sensor had 32 strain gages, whereas the arrangement only required 24. 

We made changes to two bridges of the 24 strain gage arrangement, *S*_FZ_ and *S*_MZ_. We used a simulation to establish whether there was any improvement in overall crosstalk analysis, but there was none except on *S*_FZ_ and *S*_MZ_. We modified the sensor based on the 32 strain gage arrangement. We simulated and tested the sensor. The result was vastly superior. Using a 32 strain gage is significantly superior to using a 24 strain gage, in terms of both performance and the cost.

However, the process of collecting data with a NI-DAQ is costly and prone to voltage drops that cause inaccuracy. To tackle this problem, we proposed the incorporation of a DAQ within the sensor. We created a DAQ of a small size to keep the sensor compact. The advantage of having an integrated DAQ, other than improving accuracy and crosstalk, is that it could be installed with the decoupling method. This signifies that the sensor will directly output force and moment values, not voltage values. First, we calibrated the sensor to obtain a linear equation by using LSM. Second, we installed the linear equation to the integrated DAQ. This procedure cannot be carried out using external DAQ. The previous process of converting voltage readings into force/moment was carried out in DAQ’s software. We put the proposed integrated DAQ to the test and compared the results with the external DAQ. We concluded that the integrated DAQ was successful because it met the performance and financial criteria. 

To comprehend the significance of our outcome, we compared it to the most recent results with the best 2% measurement error and crosstalk as the benchmark [20,22,26,27], as shown on Table 12. Our comprehensive design of the six-axis F/M sensor generated the best results in the strain gage type F/M sensor category, with similar specifications. 

## 7. Conclusions

This paper describes the development of a low-cost and accurate six-axis F/M sensor, which is best described in terms of the following characteristics: The structural design was updated through an optimization process using the SQP method on Timoshenko’s strain equation and validated by FEA. The maximum difference of the strain calculation of these methods on the optimized design is 2.27%.The revolutionary strain gage arrangement was chosen based on a comparison of two arrangements through simulation and experiment. Comparison through simulation shows that both arrangements have the same maximum crosstalk of −0.34%. Comparison through experiments show that the 24 strain gage arrangement has the maximum average error on *S*_MZ_, of 2.40%, and the maximum crosstalk on *S*_MZ_, of 3.35%. The 32 strain gage arrangement has the maximum average error on *S*_MY_, of 1.13%, and the maximum crosstalk on *S*_MX_, of 1.20%. These numbers make the 32 strain gage arrangement preferable.The integrated DAQ has been designed to fit inside the compact six-axis F/M sensor. The results are satisfactory and superior to commercial DAQs. According to the findings, the maximum average error and the maximum crosstalk have been reduced to 1.15% and 0.68%, respectively.

This research could serve as a guideline for comprehensive processes of building a six-axis F/M sensor and could function as a piece of foundational knowledge in the design of force sensors of any type. Nevertheless, this study is limited by its low sampling rate. It is suggested that in future work, higher MCU and ADC specifications be used to increase the sampling rate. 

## Figures and Tables

**Figure 1 sensors-21-04498-f001:**
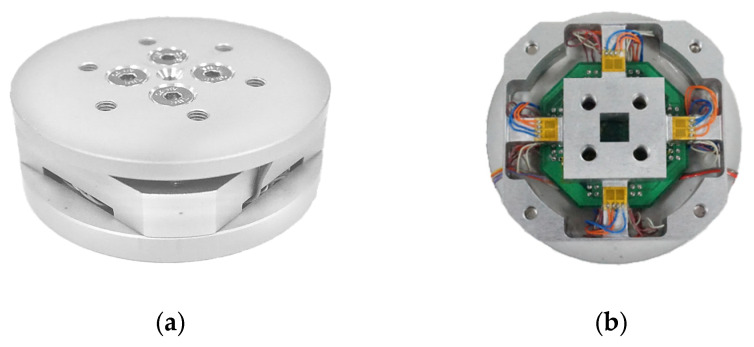
(**a**) Six-Axis F/M sensor assembled. (**b**) Elastic body of six-axis F/M sensor.

**Figure 2 sensors-21-04498-f002:**
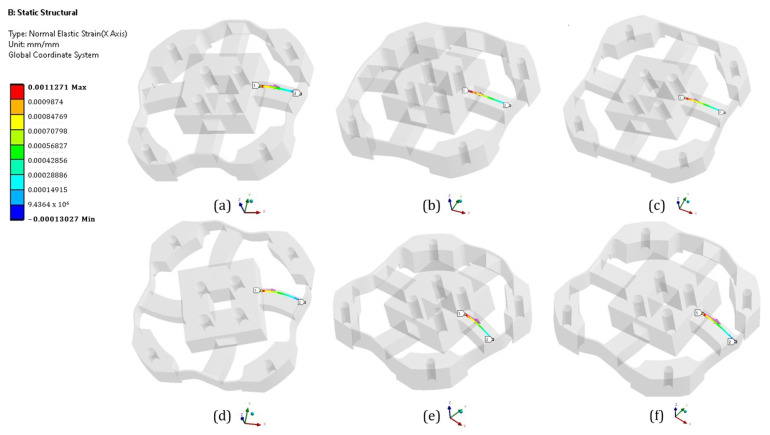
Normal strain (*X*-axis) analysis in the middle line based on acting force (**a**) F_Y_ loading on middle side line, (**b**) F_Z_ loading middle top line, (**c**) M_Y_ loading on middle top line, (**d**) M_Z_ loading on middle side line, (**e**) Multiple loading on middle top line, and (**f**) multiple loading on middle side line.

**Figure 3 sensors-21-04498-f003:**
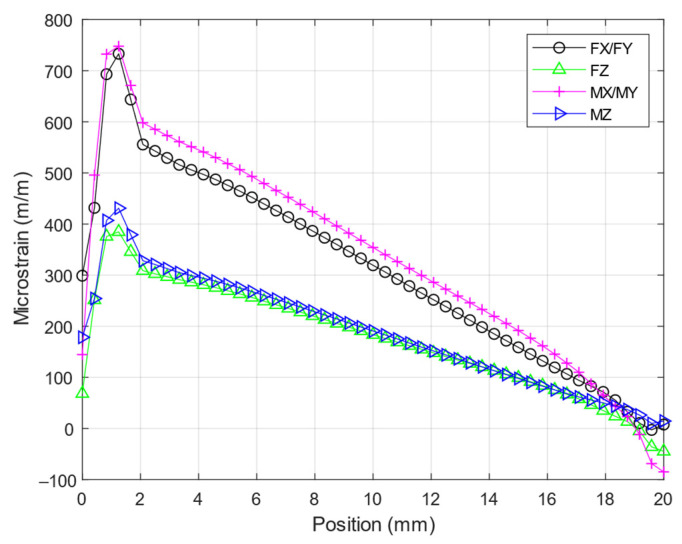
Normal strain analysis on the beam of the elastic body.

**Figure 4 sensors-21-04498-f004:**
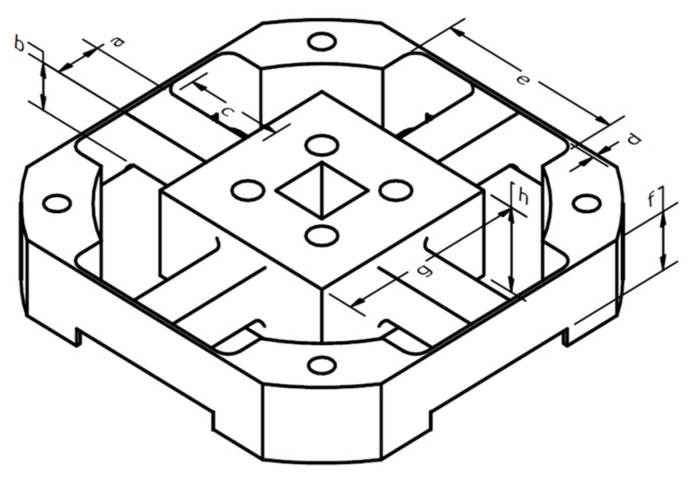
The initial design of the elastic body.

**Figure 5 sensors-21-04498-f005:**
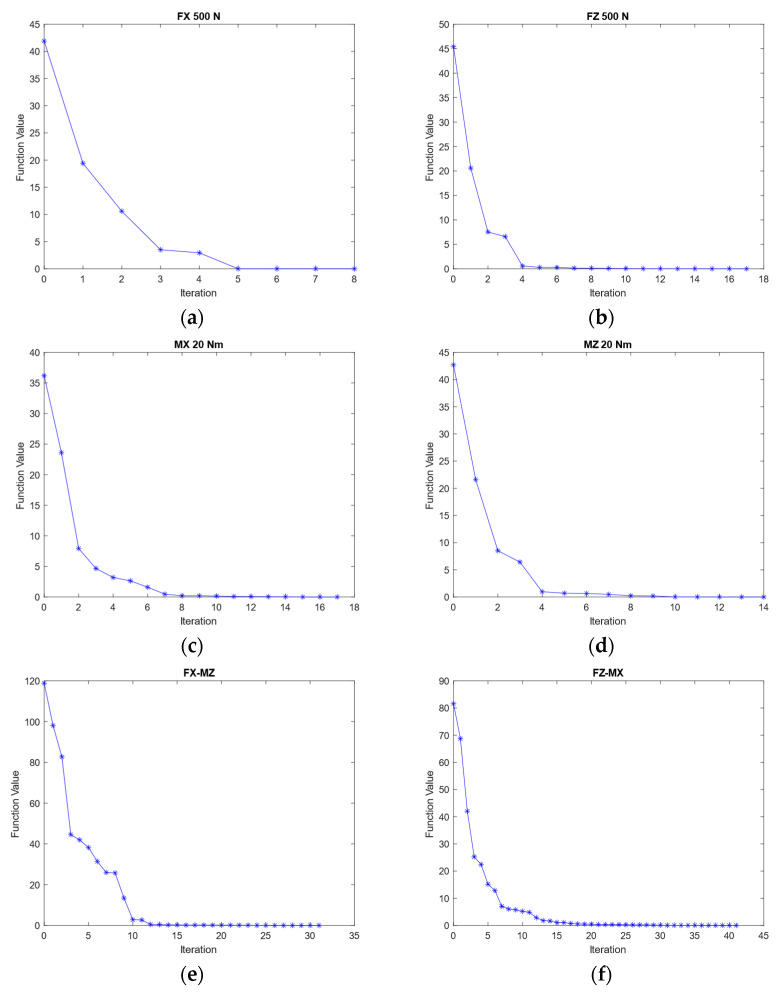
Function value of optimization during different pure load (**a**) F_X_ 500 N, (**b**) F_Z_ 500 N, (**c**) M_X_ 20 Nm, and (**d**) M_Z_ 20 Nm, and during multiple loads (**e**) F_X_ 500 N and M_Z_ 20 Nm, and (**f**) F_Z_ 500 N and M_X_ 20 Nm.

**Figure 6 sensors-21-04498-f006:**
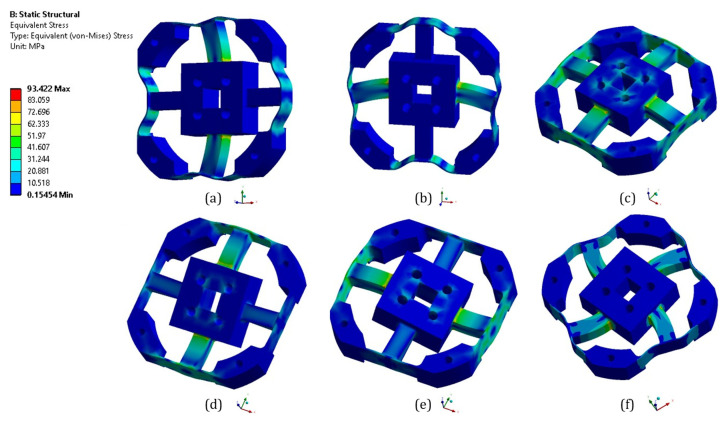
Equivalent stress from finite element analysis from different pure loading (**a**) F_X_ = 500 N, (**b**) F_Y_ = 500 N, (**c**) F_Z_ = 500 N, (**d**) M_X_ = 20 Nm, (**e**) M_Y_ = 20 Nm, and (**f**) M_Z_ = 20 Nm.

**Figure 7 sensors-21-04498-f007:**
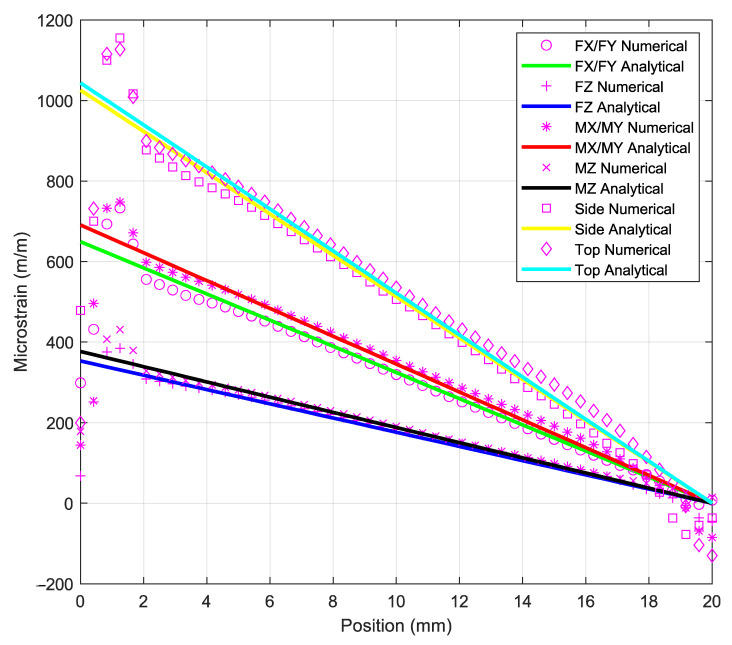
Normal strain analysis on the beam of the elastic body: numerical vs. analytical.

**Figure 8 sensors-21-04498-f008:**
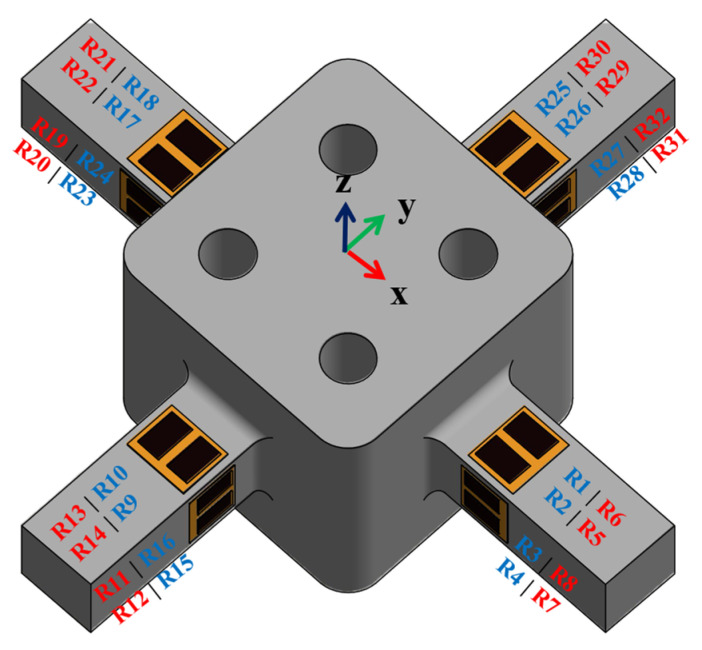
Strain gage placement on the structure.

**Figure 9 sensors-21-04498-f009:**
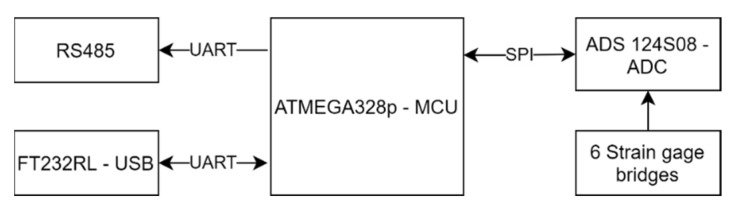
Diagram block of integrated DAQ.

**Figure 10 sensors-21-04498-f010:**
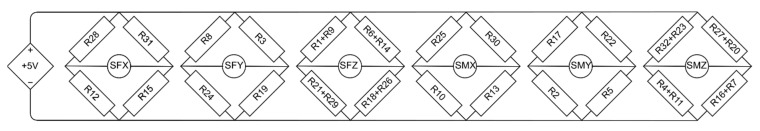
Six Wheatstone bridges circuit.

**Figure 11 sensors-21-04498-f011:**
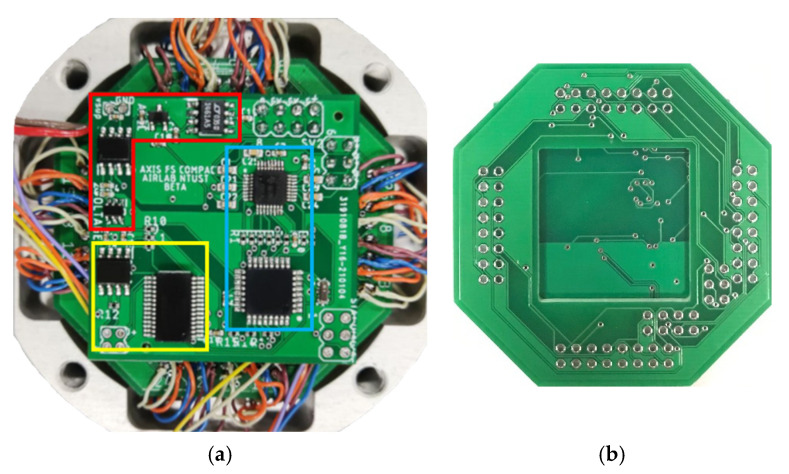
PCB layout of integrated DAQ (**a**) front (**b**) back.

**Figure 12 sensors-21-04498-f012:**
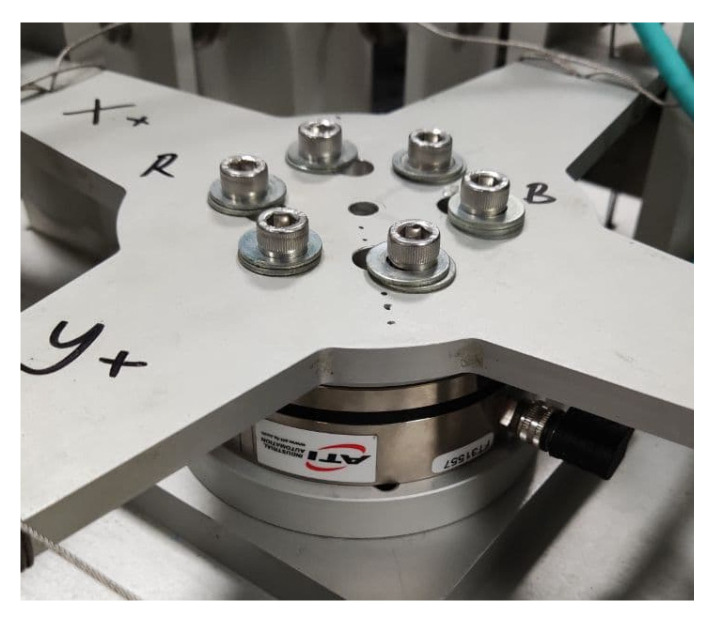
ATI AXIA 80 validated calibration jig.

**Figure 13 sensors-21-04498-f013:**
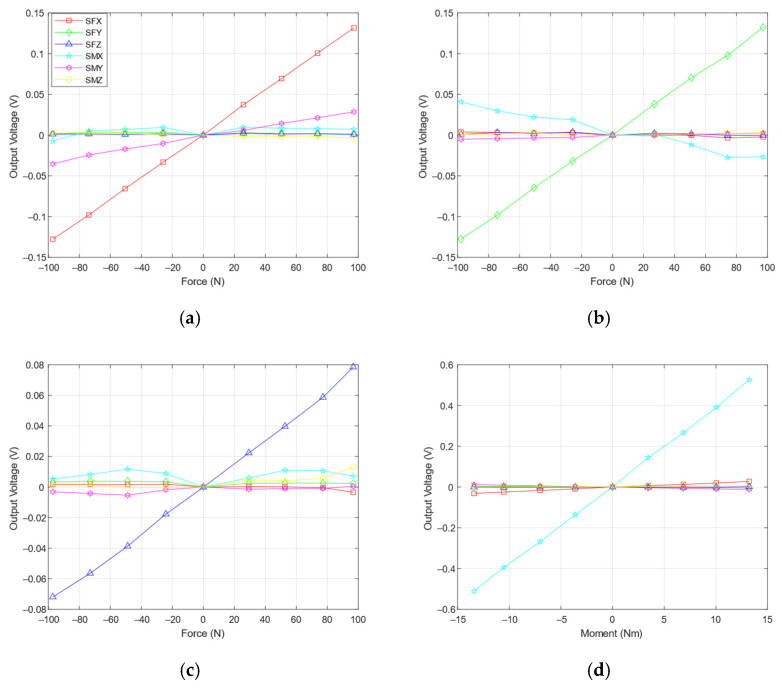
Analog output voltages (**a**) F_X_, (**b**) F_Y_, (**c**) F_Z_, (**d**) M_X_, (**e**) M_Y_, and (**f**) M_Z_.

**Table 1 sensors-21-04498-t001:** Structural design specification.

Parameters	Value	Unit
Diameter	91	mm
Height	31	mm
Weight	334	g
Force range (F_X_, F_Y_, F_Z_)	±500	N
Moment range (M_X_, M_Y_, M_Z_)	±20	Nm

**Table 2 sensors-21-04498-t002:** Initial designs of parameters.

Parameter	Dimension (mm)
Length of the elastic beam	20.0
Height of the elastic beam	8.0
Width of the elastic beam	8.3
Width of the square platform	35.0
Height of square platform	30.0
Width of elastic boundary	0.7
Height of elastic boundary	22.0
Length of elastic boundary	35.0

**Table 3 sensors-21-04498-t003:** Result of optimization with different pure loads and multiple loads.

Parameter	F_X_ (mm)	F_Z_ (mm)	M_X_ (mm)	M_Z_ (mm)	Fx+Mz (mm)	Fz+Mx (mm)	Average (mm)	Prototype (mm)
Length of the elastic beam	19.9	19.0	20.0	19.9	19.1	19.5	19.6	20.0
Height of the elastic beam	8.6	8.6	8.5	8.7	8.1	8.6	8.5	8.4
Width of the elastic beam	8.0	8.2	8.8	8.0	8.7	8.4	8.4	8.4
Width of the square platform	35.0	35.0	32.0	35.0	33.5	33.8	34.1	35.0
Height of square platform	15.0	15.0	15.0	15.0	15.0	15.0	15.0	15.0
Width of elastic boundary	0.8	0.8	0.8	0.8	0.8	0.9	0.8	0.8
Height of elastic boundary	10.8	15.0	10.0	15.0	11.4	14.9	12.9	13.4
Length of elastic boundary	32.5	35.0	35.0	35.0	33.5	30.9	33.7	34.5

**Table 4 sensors-21-04498-t004:** Normal strain analysis on the beam of the elastic body: numerical vs. analytical on 5 mm.

	Numerical Simulation	Analytical Solution	Errors
F_X_	476	487	2.27%
F_Y_	476	487	2.27%
F_Z_	269	265	−1.79%
M_X_	518	518	−0.01%
M_Y_	518	518	−0.01%
M_Z_	282	282	−0.24%
Side	769	752	−2.27%
Top	782	786	0.39%

**Table 5 sensors-21-04498-t005:** Crosstalk from finite element analysis of 24 strain gage arrangement.

	F_X_ 500 N		F_Y_ 500 N		F_Z_ 500 N		M_X_ 20 Nm		M_Y_ 20 Nm		M_Z_ 20 Nm		Multiple Loads
	S_ij_ (µ)	C_ij_ (%)	S_ij_ (µ)	C_ij_ (%)	S_ij_ (µ)	C_ij_ (%)	S_ij_ (µ)	C_ij_ (%)	S_ij_ (µ)	C_ij_ (%)	S_ij_ (µ)	C_ij_ (%)	S_ij_ (µ)
*S* _FX_	478	-	0	−0.03	0	0.01	0	−0.04	−2	−0.34	0	0.03	476
*S* _FY_	0	−0.07	478	-	0	0.04	−1	−0.22	0	0.00	0	−0.04	477
*S* _FZ_	0	0.01	0	0.03	270	-	−1	−0.14	0	0.04	0	0.00	270
*S* _MX_	0	−0.02	1	0.21	0	0.00	520	-	0	0.07	0	−0.01	521
*S* _MY_	−1	−0.23	0	0.00	0	−0.03	0	−0.01	520	-	0	0.02	519
*S* _MZ_	0	−0.04	0	0.04	0	−0.04	0	−0.01	0	−0.03	283	-	283

**Table 6 sensors-21-04498-t006:** Crosstalk from finite element analysis of 32 strain gage arrangement.

	F_X_ 500 N		F_Y_ 500 N		F_Z_ 500 N		M_X_ 20 Nm		M_Y_ 20 Nm		M_Z_ 20 Nm		Multiple Loads
	S_ij_ (µ)	C_ij_ (%)	S_ij_ (µ)	C_ij_ (%)	S_ij_ (µ)	C_ij_ (%)	S_ij_ (µ)	C_ij_ (%)	S_ij_ (µ)	C_ij_ (%)	S_ij_ (µ)	C_ij_ (%)	S_ij_ (µ)
*S* _FX_	478	-	0	−0.03	0	0.01	0	−0.04	−2	−0.34	0	0.03	476
*S* _FY_	0	−0.07	478	-	0	0.04	−1	−0.22	0	0.00	0	−0.04	477
*S* _FZ_	0	0.02	0	0.02	270	-	0	−0.06	1	0.13	0	−0.02	271
*S* _MX_	0	−0.02	1	0.21	0	0.00	520	-	0	0.07	0	−0.01	521
*S* _MY_	−1	−0.23	0	0.00	0	−0.03	0	−0.01	520	-	0	0.02	519
*S* _MZ_	0	0.00	0	0.01	0	−0.02	0	−0.01	0	−0.03	283	-	283

**Table 7 sensors-21-04498-t007:** Electrical specification of the six-axis F/M sensor.

Parameters	Value	Unit
Resolution (F_X_, F_Y_, F_Z_)	± 0.1	N
Resolution (M_X_, M_Y_, M_Z_)	± 0.001	Nm
Sampling Rate	100	Hz

**Table 8 sensors-21-04498-t008:** Average force and moment reading of ATI AXIA.

No	LOAD F_X_	LOAD F_Y_	LOAD F_Z_	LOAD M_X_	LOAD M_Y_	LOAD M_Z_
1	97.15	97.12	96.71	13.22	12.97	9.90
2	73.72	74.20	77.18	10.03	10.48	7.57
3	50.45	50.80	52.65	6.85	7.08	5.30
4	25.87	26.97	29.34	3.45	3.62	2.62
5	−25.96	−25.79	−24.19	−3.62	−3.26	−2.53
6	−50.49	−50.70	−48.85	−7.02	−6.78	−5.01
7	−73.87	−74.38	−73.02	−10.52	−9.96	−7.59
8	−97.15	−97.82	−97.07	−13.40	−13.05	−9.75

**Table 10 sensors-21-04498-t010:** Analysis of average measurement error and crosstalk of 32 strain gage arrangement.

	LOAD F_X_	LOAD F_Y_	LOAD F_Z_	LOAD M_X_	LOAD M_Y_	LOAD M_Z_
*S* _FX_	0.27%	0.18%	0.24%	0.17%	0.41%	0.33%
*S* _FY_	0.20%	0.30%	0.28%	0.21%	0.21%	0.31%
*S* _FZ_	0.30%	0.43%	0.76%	0.18%	0.30%	0.43%
*S* _MX_	0.22%	0.37%	0.34%	0.68%	1.20%	0.58%
*S* _MY_	0.16%	0.15%	0.33%	0.12%	1.13%	0.17%
*S* _MZ_	0.25%	0.49%	0.92%	0.89%	0.62%	0.62%

**Table 11 sensors-21-04498-t011:** Analysis of average measurement error and crosstalk of 32 strain gage arrangement with integrated DAQ.

	LOAD F_X_	LOAD F_Y_	LOAD F_Z_	LOAD M_X_	LOAD M_Y_	LOAD M_Z_
*S* _FX_	0.11%	0.09%	0.17%	0.15%	0.32%	0.25%
*S* _FY_	0.09%	0.11%	0.08%	0.43%	0.16%	0.14%
*S* _FZ_	0.47%	0.57%	0.30%	0.46%	0.58%	0.43%
*S* _MX_	0.08%	0.07%	0.36%	0.64%	0.47%	0.07%
*S* _MY_	0.20%	0.27%	0.16%	0.40%	1.15%	0.15%
*S* _MZ_	0.46%	0.60%	0.42%	0.31%	0.68%	0.68%

**Table 12 sensors-21-04498-t012:** The comparison on maximum measurement error and crosstalk.

	Proposed	[20]	[26]	[22]	[27]
F_X_, F_Y_, F_Z_	500 N, 500 N, 500 N	400 N, 400 N, 800 N	400 N, 400 N, 800 N	400 N, 400 N, 800 N	400 N, 400 N, 1000 N
M_X_, M_Y_, M_Z_	20 Nm, 20 Nm, 20 Nm	30 Nm, 30 Nm, 30 Nm	40 Nm, 40 Nm, 40 Nm	40 Nm, 40 Nm, 40 Nm	20 Nm, 20 Nm, 10 Nm
Measurement Error	1.15%	2.00%	5.20%	1.78%	1.88%
Crosstalk	0.68%	2.63%	−3.20%	4.78%	3.00%

**Table 9 sensors-21-04498-t009:** Analysis of average measurement error and crosstalk of 24 strain gage arrangement.

	LOAD F_X_	LOAD F_Y_	LOAD F_Z_	LOAD M_X_	LOAD M_Y_	LOAD M_Z_
*S* _FX_	0.42%	0.19%	0.37%	0.31%	0.23%	0.38%
*S* _FY_	0.18%	0.19%	0.35%	0.32%	0.28%	0.17%
*S* _FZ_	0.33%	0.23%	0.78%	0.76%	1.02%	0.51%
*S* _MX_	0.29%	0.21%	0.98%	1.00%	0.39%	0.16%
*S* _MY_	0.05%	0.08%	0.09%	0.21%	1.06%	0.27%
*S* _MZ_	1.61%	1.04%	3.24%	3.35%	2.40%	2.40%

## Data Availability

The data presented in this study are available on request from the corresponding author. The data are not publicly available due to privacy.

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
