# Peer review of "A Comprehensive Design of Six-Axis Force/Moment Sensor"

_sensors, 2021, doi:10.3390/s21134498_

Round 1

Reviewer 1 Report

  1. In Fig. 2, the microstrain direction and position line of the beam should be explained. Also, in Fig. 6, why is there non-linear strains in the final tail of numerical results?
  2. How to determine the specific strain limits for each axis?
  3. The strain distribution formula (1)~(4) is only for the condition of a single load. When loads in six directions act together, the formula should be different, and the strain may exceed the maximum allowable strain of 500, this problem should be taken into account during optimization.
  4. The data in Fig. 6 do not match those in Table 4.
  5. Why are two single strain gages in parallel used? What’s the advantage? In addition, the final size and strain distribution of the elastic body after optimization, and the arrangements of 16 and 32 strain gauges should be displayed clearly, and the principle of decoupling should be explained.
  6. In table 5 and table 6, how to calculate the value of ? Why is the maximum crosstalk in table 5 same as that in table 6? What’s the different advantage and disadvantage of 24-strain gages and 24-strain gages arrangements?
  7. In Section 3, the sensor's bridge conditioning circuit and the range of its analog signal output should be displayed.
  8. How to determine the resolutions of six-axis forces in table 7?
  9. What’s the absolute average errors in tables 9, 10 and 11? What values are the crosstalks in these tables?
  10. How much are the temperature coefficients of the sensor in the specific temperature range?

Author Response

Firstly, we would like to thank the reviewer for his/her in-depth and careful reading of our manuscript and valuable suggestions. We have revised the manuscript by following the reviewer’s advices as much as possible. We have tried our best to provide all suitable answers for the questions according to our knowledge. Please see the attachment.

Reviewer 2 Report

Dear Authors,

The manuscript is very interesting, the authors have done a good job of reviewing the literature, describing the subject and the scope of the research.
It should be emphasized that I conducted my own research supported by simulations used Matlab. Overall work is good I only have some minor comments below:

  1. You can change the first keyword to avoid repeating this phrase in the title. (Please consider it).
  2. in line 35 there is the error it should be rather "external".
  3. in lines 68-72 you write the type of material the sensor is made of, is it only aluminum or maybe there are other, e.g. plastics?
  4. in line 108 is the error do you have a period and a comma there?
  5. in line 146 the period after the square bracket is missing.
  6. in Table 4, in second line is error - double dot.
  7. In lines 143-146 the authors write that they obtained better results in relation to previous studies, but did not write about what this improvement could result from. Perhaps it is worth discussing this issue.
  8. In the 297-300 you write that your sensor produced the best results in the F / M strain gauge category with similar specifications, whether they were commercial sensors or those developed by other research teams. No examples are given in the paper.

Thank you and good luck.

Author Response

(The authors gave the same response as above.)

Round 2

Reviewer 1 Report

  1. When analyzing the stress distributions under multiple loads, why only consider two conditions: multiple loads of Fx and Mz, multiple loads of Fz and Mx, why not analyzing the stress under the loads of Fx, Fy and Mz, the loads of Fz, Mx and My, and more universally, the stress under the loads of six directions. For example, the stress contribution of Fy to the middle side line of the beam under Fx may be small, but it should be analyzed, which is important to analyze the reasons of cross-axis sensitivity.
  2. In Figure 12 (a), the cross-axis sensitivity of SMy to Fx, and in Figure 12(b), the cross-axis sensitivity of SMx to Fy are very obvious. The reasons for these results can be further analyzed and discussed.
  3. In Figure 4 and Figure 9, it would be better to attach a schematic diagram of the arrangement of each strain gauge on the elastic body.
  4. In Figures 2 and Figure 6, the deformation and stress distributions of the elastic body are not easy to be observed, changing the angle of view may be better.

Author Response

(The authors gave the same response as above.)
